# Circulation responses to surface heating and implications for polar amplification

Peter Yu Feng Siew[1], Camille Li[2,3,1], Stefan Pieter Sobolowski[2,3,4], Etienne Dunn-Sigouin[4,3], and Mingfang Ting[1,5]

[1]Lamont-Doherty Earth Observatory, Columbia University, Palisades, NY, USA
[2]Geophysical Institute, University of Bergen, Bergen, Norway
[3]Bjerknes Centre for Climate Research, Bergen, Norway
[4]NORCE, Bergen, Norway
[5]Columbia Climate School, Columbia University, New York, NY, USA

**Correspondence:** Peter Yu Feng Siew (pyfsiew@ldeo.columbia.edu)

**Abstract.** A seminal study by Hoskins and Karoly (1981) explored the atmospheric circulation response to tropospheric heating perturbations at low and mid latitudes. Here we revisit and extend their study by investigating the circulation and temperature response to low, mid and high latitude surface heating using an idealised moist, gray radiation model. Our results corroborate previous findings showing that heating perturbations at low and mid latitudes are balanced by different time-mean circulation responses - upward motion and horizontal temperature advection, respectively. Transient eddy heat flux divergence plays an increasingly important role with latitude, becoming the main circulation response at high latitudes. However, this mechanism is less efficient at balancing heating perturbations than temperature advection, leading to greater reliance on an additional contribution from radiative cooling. These dynamical and radiative adjustments promote stronger lower tropospheric warming in response to surface heating at high latitudes compared with lower latitudes. This elucidates the mechanisms by which sea ice loss contributes to polar amplification in a warming climate.

## 1 Introduction

There is strong consensus that the Arctic has warmed at a rate more than twice the global average over recent decades (e.g., Serreze et al., 2009; Cohen et al., 2014; Previdi et al., 2021; England et al., 2021; Rantanen et al., 2022). Concurrent with this warming, there has been a rapid decline in sea ice that acts as both a response and a contributor to the Arctic amplification signal (Cohen et al., 2014; Dai et al., 2019; Olonscheck et al., 2019). The shrinking sea ice cover exposes relatively warm ocean waters that warm the lower troposphere through turbulent exchanges. It is hypothesised that this additional energy input at the surface can modify the large-scale circulation through several proposed mechanisms (see Cohen et al., 2020; Outten et al., 2022). However, detecting such linkages and identifying robust physical mechanisms poses a challenge (e.g., Sellevold et al., 2016; Shepherd, 2016; Mori et al., 2019; Siew et al., 2020, 2021; Labe et al., 2020; Sun et al., 2022; Shaw and Smith, 2022; Zheng et al., 2023), in large part because our dynamical understanding of Arctic-midlatitude teleconnections remains incomplete (Wallace et al., 2014; Hoskins and Woollings, 2015; Woollings et al., 2023).

There is a rich history of investigations aimed at improving fundamental understanding of the atmospheric response to heating perturbations. A paradigmatic demonstration of differences in how the atmosphere adjusts to heating in the tropics versus the midlatitudes was presented by Hoskins and Karoly (1981). Using a linearized, five-layer baroclinic model, Hoskins and Karoly (1981) showed that the circulation response to heating attempts to offset or balance the perturbation in the most efficient way possible. In the tropics, near-surface atmospheric heating induces deep tropospheric warming, and the resulting upward motion produces adiabatic cooling to balance the extra energy input. Conversely, in the midlatitudes, vertical motion is inhibited and near-surface heating is mainly balanced by horizontal temperature advection. In the lower troposphere, meridional advection is induced by a low-pressure anomaly downstream of the heating, which brings cold air from the pole; zonal advection plays an important role in the free troposphere, where zonal winds are stronger. Apart from these local responses, the surface heating, especially in the tropics, can excite large-scale Rossby waves which redistribute the energy input associated with anomalous heating. Similar dynamical responses to tropical and midlatitude surface heating have also been found in more comprehensive models and setups (Ting and Held, 1990; Ting, 1991; Hall et al., 2001; Walter et al., 2001; Inatsu et al., 2002, 2003; Deser et al., 2007).

Given the tropospheric heating associated with Arctic amplification of global warming and the fact that the circulation response to said heating is ambiguous, it is natural to extend the study of heating perturbations to high latitudes. At high latitudes, the high static stability, weak zonal wind, and weak meridional temperature gradient limit the ability of vertical and horizontal temperature advection to balance heating perturbations by the aforementioned atmospheric processes (Woollings et al., 2023). Additional processes such as radiative cooling have been hypothesised to be important in the polar regions (Kim et al., 2021; Woollings et al., 2023; Miyawaki et al., 2023), but the full adjustment mechanism has not been systematically explored. Such an investigation also offers an opportunity to revisit and extend the tropical and midlatitude heating results from Hoskins and Karoly (1981) using a gray radiation aquaplanet model that has no clouds or ice (Frierson et al., 2006). This modelling framework is still idealised, but allows us to include some effects of moisture, surface energy fluxes and radiation that are hypothesised to help set the temperature and circulation responses to high-latitude heating (e.g., Winton, 2006; Langen et al., 2012; Kim and Kim, 2017; Matthews et al., 2022).

In this study, we carry out perturbation experiments by imposing surface heating from low (15°N) to high (75°N) latitudes in a zonally symmetric climate. The aforementioned idealised framework is well suited to isolating the fundamental physical processes that drive the atmospheric response to prescribed heating perturbations. We examine the physical links between atmospheric circulation and temperature responses to the heating perturbations, focusing on the differences between the polar heating experiments and heating at other latitudes.

## 2 Methods

### 2.1 Model setup and control experiment

We employed the idealised moist general circulation model documented in Frierson et al. (2006); Frierson (2007); O'Gorman and Schneider (2008). The model was run using the ISCA idealised modelling framework (Vallis et al., 2018), which allows

one to easily modify the complexity of the various components in the model. In our setup, the model was integrated at T85 spectral resolution in the horizontal (1.4°x1.4° grid size) with 30 unevenly spaced sigma levels, from the surface to 3 Pa, in the vertical.

The model consists of a primitive equation atmosphere coupled to a slab ocean. The slab ocean does not include sea ice or the effects of changes in ocean circulation. Water is allowed to evaporate from the ocean to form water vapour, which is advected with the atmospheric flow. Moisture interacts with the atmospheric dynamics via the release of latent heat during condensation. All condensed water is precipitated immediately, hence there are no clouds in the model. Additionally, the model uses a gray radiation scheme, where a single optical thickness is used across the entire longwave frequency band, meaning that radiative water vapour feedbacks are not represented. The longwave and shortwave optical thicknesses are prescribed as a function of latitude and altitude.

The control experiment follows the protocol of the "Tropical Rain belts with an Annual Cycle and continent Model Inter-comparison Project" (TRACMIP), which has been used for aquaplanet setups in previous modelling studies (e.g., Voigt et al., 2016; Dunn-Sigouin et al., 2021). The protocol includes insolation forcing with present-day diurnal and seasonal cycles, and a 30-metre slab ocean with a time-independent and zonally symmetric ocean heat flux convergence (Q-flux) that mimics the observed equator-to-pole ocean heat transport (see Equation 3 in Voigt et al., 2016). All other parameters not specified in the TRACMIP protocol were set following O'Gorman and Schneider (2008). The model was spun up for 20 years, and then run for an additional 30 years to produce the control experiment. The zonal-mean zonal wind, air temperature and transient eddy kinetic energy of the control experiment are shown in Figure S1. The temperature and wind show reasonable agreement with observations, although the eddies in the upper troposphere appear to be weaker in the model. In the control simulation, the weaker eddies may stem from the model's absence of sea ice (Miyawaki et al., 2023). However, our findings remain consistent across the winter and summer seasons, which respectively exhibit stronger and weaker eddies compared to the annual mean, suggesting that our results are not sensitive to eddy strength in the control simulation.

## 2.2 Perturbation experiments

Experiments with imposed, time-independent Q-flux perturbations representing localised heating anomalies were branched off from the control experiment. Each perturbation experiment was run for 30 years after discarding a 10-year spin-up. The system reaches equilibrium around the middle of this spin-up period, evident from the stabilisation of global mean surface temperature (Figure S2). The form of the Q-flux perturbation follows the analytical formula in Thomson and Vallis (2018):

$$Q_{max} * \{1 - [(\frac{\phi - \phi_o}{\phi_d})^2 + (\frac{\lambda - \lambda_o}{\lambda_d})^2]]\} \tag{1}$$

The Q-flux perturbation has a maximum magnitude of $Q_{max}$ at the central latitude and longitude of $\phi_o$ and $\lambda_o$, respectively. $\phi_d$ and $\lambda_d$ describe the tapering of the Q-flux perturbation away from the maximum. We have five Q-flux perturbation experiments with $\phi_o$ = 15°N, 30°N, 45°N, 60°N and 75°N spanning $\lambda_d$ = 32°, 36°, 44°, 62° and 120° of longitude, respectively. For all experiments, we set the heating perturbation at $\lambda_o$ = 0° spanning $\phi_d$ = 30° of latitude, and set $Q_{max}$ = 100Wm$^{-2}$ (see Figure 1). By design, all Q-flux perturbations are identical in areal extent and input the same amount of total energy into the system

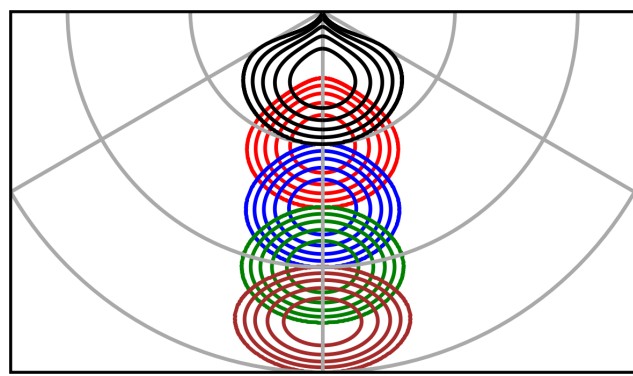

**Figure 1.** Prescribed Q-flux in the heating perturbation experiments with heating at 15°N (brown), 30°N (green), 45°N (blue), 60°N (red) and 75°N (black). The latitude lines mark 0°N, 30°N and 60°N. The longitude lines mark 60°W, 0° and 60°E. Contour intervals are 0, 20, 40, 60 and 80 Wm$^{-2}$. All Q-flux perturbations are identical in areal extent and input the same amount of total energy into the system (about 450TW).

(about 450TW). The response of a variable to the Q-flux perturbation is obtained by subtracting the annual-mean climatology of the control run from the perturbation experiment (i.e., perturbation minus control).

## 2.3 Thermodynamic equation

Following the framework from Hoskins and Karoly (1981), we use the steady-state thermodynamic equation to diagnose which heat transport terms balance the diabatic heating generated by the surface Q-flux perturbation. We extend the linearized quasi-geostrophic form of the steady-state thermodynamic equation used in Hoskins and Karoly (1981) to the full thermodynamic equation that includes transient eddy heat fluxes (Equation 3.21 in James, 1995):

$$\overline{u}\frac{d\overline{\theta}}{dx} + \overline{v}\frac{d\overline{\theta}}{dy} + \overline{\omega}\frac{d\overline{\theta}}{dp} + \frac{d}{dx}\overline{u'\theta'} + \frac{d}{dy}\overline{v'\theta'} + \frac{d}{dp}\overline{\omega'\theta'} = \overline{Q}\left(\frac{p_o}{p}\right)^{\frac{R}{C_p}} \tag{2}$$

where $u, v, \omega$ are the zonal, meridional, vertical velocity, respectively; $\frac{d\theta}{dx}, \frac{d\theta}{dy}, \frac{d\theta}{dp}$ are the gradients of potential temperature in the zonal, meridional, vertical directions, respectively; p is pressure and $p_o$ is the reference pressure (=1000hPa); R (=287Jkg$^{-1}$K$^{-1}$) is the gas constant of air; and $C_p$ (=1004Jkg$^{-1}$K$^{-1}$) is the specific heat capacity of air at a constant pressure. The bars represent the time-mean and primes represent the deviation from the time-mean. The first three terms on the left side of Equation 2 are the time-mean advection of potential temperature in the zonal, meridional and vertical directions. The sum of the last three terms on the left side is the transient eddy heat flux divergence. On the right side, Q is the diabatic heating rate. The $\overline{Q}$ calculated from Equation 2 as a residual of the transport terms on the left side is compared with the diabatic heating calculated from source terms in a column-integrated sense, and they yield very similar results (Figure S3). This confirms that the residual method provides a good estimate of the diabatic heating.

## 2.4 Moist static energy budget

The thermodynamic equation (Equation 2) quantifies the contributions of various transport terms (circulation processes) for balancing the diabatic heating added by the Q-flux perturbation. To bring in the relative importance of radiative processes, we consider the steady-state column-integrated moist static energy budget (similar to Equation 13.47 in Peixoto and Oort, 1992):

$$F_{surface} = F_{top} + \nabla \cdot \overline{\mathbf{u}} E \qquad (3)$$

where $F_{surface}$ describes the surface turbulent (sensible and latent) and radiative fluxes, $F_{top}$ describes the top-of-the-atmosphere radiative fluxes, $\overline{\mathbf{u}}$ describes the vertical integral of zonal and meridional winds, and E describes the vertical integral of moist static energy. The last term ($\nabla \cdot \overline{\mathbf{u}} E$) describes the transport of the vertical integral of moist static energy, and is treated as residual of the first and second terms. Note that the transport term here represents the total transport (both time-mean and transient terms) of the moist static energy (the sensible, latent, geopotential and kinetic energy) in a column-integrated sense, which is different from the transport of potential temperature decomposed into time-mean and transient terms level by level on the left side of Equation 2.

## 3 Results

### 3.1 Temperature responses to heating perturbations from low to high latitudes

In this section, we explore the atmospheric response to surface heating from low to high latitudes via the previously described imposed Q-flux perturbation. The first-order response to surface heating is increased temperature at the surface (Figure 2) and throughout the troposphere (Figure 3) collocated with the heating sources. As the heating is moved from 15°N to 75°N, the surface warming increases in magnitude (Figure 2), and the maximum warming shifts from the upper to the lower troposphere (Figure 3). The additional stratospheric warming for the 75°N heating experiment is consistent with an eddy-driven adjustment involving anomalous subsidence (see Hell et al., 2020). This response, while important, is not the focus of the present study. The amplified temperature response in the polar heating experiment resembles the Arctic amplification signal observed over the past few decades, whereby the Arctic has warmed faster than other parts of the globe, especially near the surface and in regions of rapid sea ice loss (Cohen et al., 2020; England et al., 2021; Rantanen et al., 2022).

We will first employ the idealised framework established in Hoskins and Karoly (1981) to explore what circulation responses balance the diabatic heating across the heating perturbation experiments (sections 3.2 and 3.3). Given that radiative processes are also important, we next quantify the relative importance of the circulation and radiative processes in balancing the energy input from the heating perturbations (section 3.4). We start with heating perturbations centred at low and mid latitudes (15°N, 30°N and 45°N), which yield results consistent with the near-surface heating experiments in Hoskins and Karoly (1981). Using these experiments as a baseline, we then move the heating perturbation towards higher latitudes (60°N and 75°N), highlighting differences in the responses that can help clarify the physical link between real-world sea ice loss and Arctic amplification of global warming.

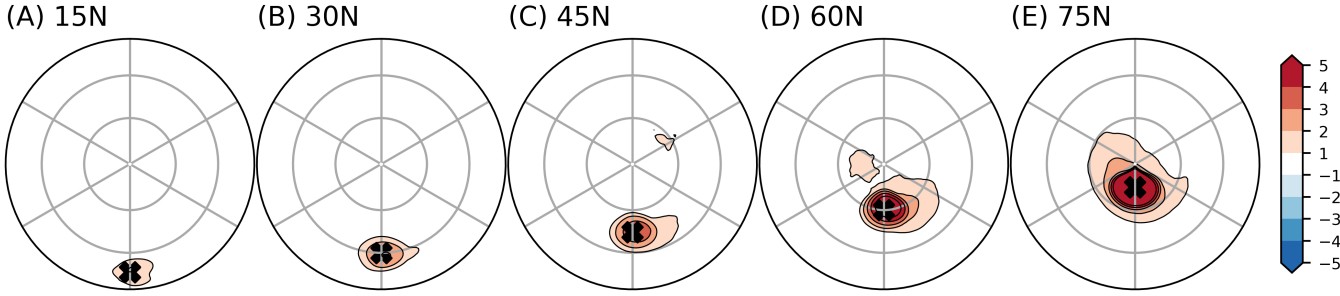

**Figure 2.** Surface temperature response (K) in the **(A)** 15°N, **(B)** 30°N, **(C)** 45°N, **(D)** 60°N and **(E)** 75°N heating perturbation experiments. The crosses mark the position of the surface heating perturbation. The latitude lines mark 30°N and 60°N. The longitude lines denote 60° intervals, marking 120°W, 60°W, 0°, 60°E, 120°E and 180°.

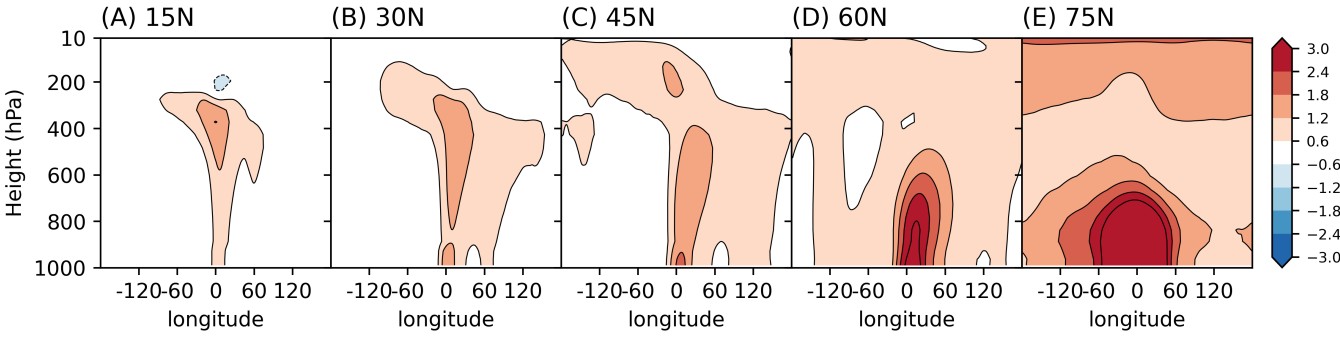

**Figure 3.** Air temperature response (K) in the **(A)** 15°N, **(B)** 30°N, **(C)** 45°N, **(D)** 60°N and **(E)** 75°N heating perturbation experiments. The longitude-height section shown is a meridional average over ±30° of latitude from the central latitude of the heating perturbations.

## 3.2 Mean low-level circulation responses to heating perturbations

Low to mid-latitude surface heating generates vertical and meridional circulation responses in the lower troposphere that help balance the heating perturbation, consistent with Hoskins and Karoly (1981). The near-surface circulation response to the 15°N, 30°N and 45°N surface heating shows vertical ascent (Figure 4) and a low-pressure near the heating source (Figure 5). The
upward motion, which acts to balance the surface heating via adiabatic cooling, is particularly strong for the lowest latitude (15°N) heating perturbation. Figure S4 further shows that the vertical ascent is accompanied by strong upper-level divergence over the heating source that further moves heat away from the atmospheric column. As the heating moves from 15°N to 45°N, the anomalous upward motion becomes weaker, and the low-pressure centre shifts slightly eastwards, moving downstream of the heating source. This configuration tends to induce equatorward flow in the lower troposphere over the perturbation region
and advect cold polar air to balance the heating (Hoskins and Karoly, 1981). Apart from these local circulation responses, we also find remote stationary wave responses (which exhibit clear zonal asymmetry) at both lower (Figure 5) and upper levels

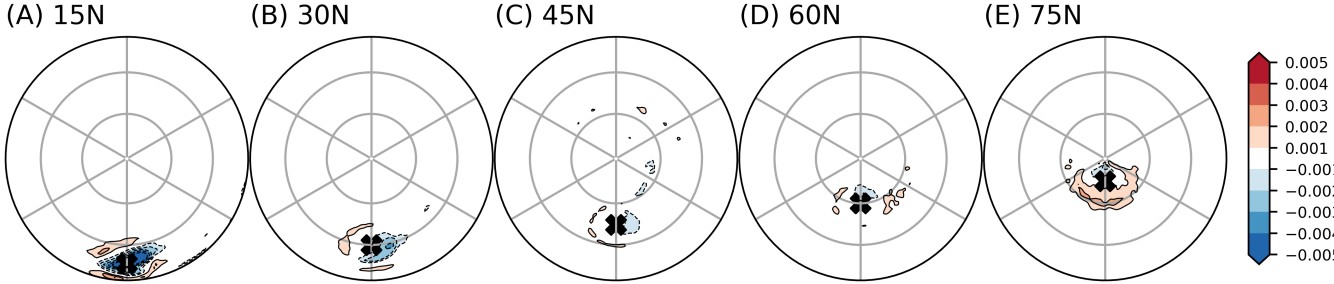

**Figure 4.** Near surface (990 hPa) vertical velocity response (Pas$^{-1}$) in the **(A)** 15°N, **(B)** 30°N, **(C)** 45°N, **(D)** 60°N and **(E)** 75°N heating experiments. The crosses mark the position of the surface heating perturbation. The latitude lines mark 30°N and 60°N. The longitude lines denote 60° intervals, marking 120°W, 60°W, 0°, 60°E, 120°E and 180°.

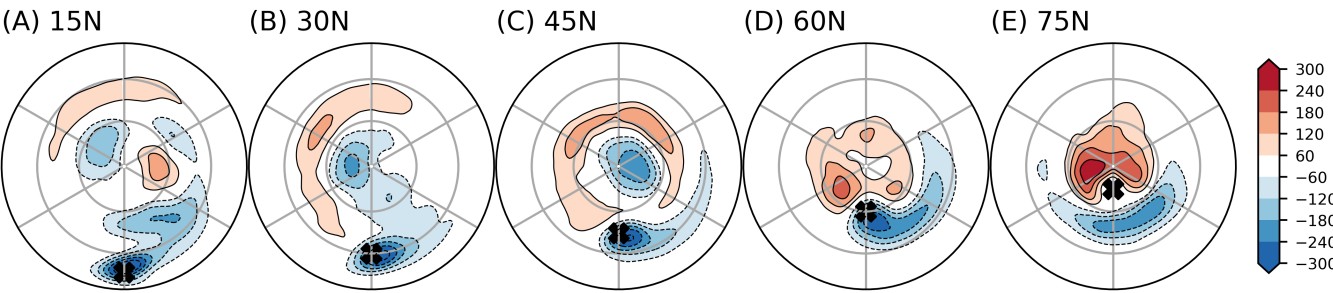

**Figure 5.** Surface pressure response (Pa) in the **(A)** 15°N, **(B)** 30°N, **(C)** 45°N, **(E)** 60°N and **(E)** 75°N heating experiments. The crosses mark the position of the surface heating perturbation. The latitude lines mark 30°N and 60°N. The longitude lines denote 60° intervals, marking 120°W, 60°W, 0°, 60°E, 120°E and 180°.

(Figure S5). The remote wave responses to heating at lower latitudes bear some similarities to those shown in Hoskins and Karoly (1981), but are not the focus of this study.

When the surface heating is placed further north at 60°N and 75°N, there is less indication that mean circulation responses in the lower troposphere work to offset the extra heat. Near-surface upward motion (adiabatic cooling) is very weak for both high-latitude heating experiments, and there is even subsidence (adiabatic warming) to the south of the perturbation in the case of the 75°N experiment (Figure 4E). The low-pressure anomaly still provides some cold-air advection to balance the 60°N heating source (Figure 5D), as in the case of the midlatitude heating experiments. However, the low shifts equatorward in the 75°N experiment (Figure 5E), and at the location of the heating source it is mostly offset by a zonally-symmetric positive anomaly at the pole, reflecting a negative Arctic Oscillation (AO) or northern annular mode (NAM) response that is commonly found in idealised polar heating (Butler et al., 2010; Wu and Smith, 2016; Zhang et al., 2018; Hell et al., 2020) and sea ice reduction (e.g., Magnusdottir et al., 2004; Deser et al., 2004, 2007, 2010) experiments. Removing the zonal-mean response to isolate the stationary wave pattern, we find a surface low nearly collocated with the 75°N heating source (not shown; consistent with

Sellevold et al., 2016). Overall, mean meridional and vertical advection does not appear to play important roles in balancing high-latitude, near-surface heating perturbations, as will be confirmed in section 3.3.

## 3.3 Role of transient eddy fluxes versus mean circulation in balancing heating perturbations

The previous section qualitatively shows how the circulation responds to near-surface heating perturbations. Next, we quantify the relative importance of the circulation processes that balance the heating according to the thermodynamic equation (Equation 2), namely, the time-mean zonal, meridional and vertical potential temperature advection, as well as the transient eddy flux divergence. Figure 6 shows these terms in height-longitude sections near the heating source for each experiment (as differences between the perturbation and control). The first row shows the diabatic heating term (right side of Equation 2). The next three rows show temperature advection by the time-mean vertical, zonal and meridional flow. The last row shows the transient eddy heat flux divergence. The sum of the heat transport terms (second to fifth row) equals the diabatic heating (first row). As described in section 2.3, a direct calculation of column-integrated diabatic heating agrees well with the residual calculation shown here.

Low-latitude (15°N and 30°N) heating perturbations create deep diabatic heating signals that maximise in the middle and upper troposphere (Figures 6A and 6B), along with weak cooling upstream of the heating maximum. Consistent with the diagnostics in the previous section, such heating is in large part balanced by upward motion and hence adiabatic cooling over the heating maximum, while the weak cooling upstream is balanced by reduced convection and hence adiabatic warming. Note that in both 15°N and 30°N heating experiments, the horizontal temperature advection terms play some role in the lower troposphere, while the role of transient eddy heat flux divergence is still negligible.

The 45°N heating perturbation generates a shallower diabatic heating profile that maximises in the low-to-mid troposphere (Figure 6C). As discussed in the previous subsection, the downstream low-pressure anomaly induces northerly flow that brings cold polar air from the north towards the western portion of the perturbation region in the lower troposphere; the mid to upper tropospheric signal is of the opposite sign, however. In the middle troposphere, zonal temperature advection also appears to play an important role in balancing the heating by advecting warm air sitting over the perturbation region further downstream, consistent with Hoskins and Karoly (1981). Apart from the time-mean horizontal advection, transient eddy heat flux divergence also moves excess heat out of the region in the low to mid troposphere, with magnitudes comparable to the meridional temperature advection near the surface. Note that the role of vertical advection is negligible throughout most of the troposphere in this experiment.

The heating perturbations at higher latitudes (60°N and 75°N) create diabatic heating profiles that are concentrated in the lower troposphere (Figures 6D and 6E). While the time-mean horizontal temperature advection still plays a role in balancing the 60°N heating in the middle and lower troposphere, the transient eddy heat flux divergence becomes as important at lower levels (Figure 6D, fifth row). For the 75°N experiment, the transient eddy heat flux divergence is the dominant response (Figure 6E).

Figure 7 summarises the results in this section by comparing the ratios of each circulation term to the total diabatic heating. The upper panel (Figure 7A) considers the main regions covered by the heating sources (averaging over ±20° of longitude

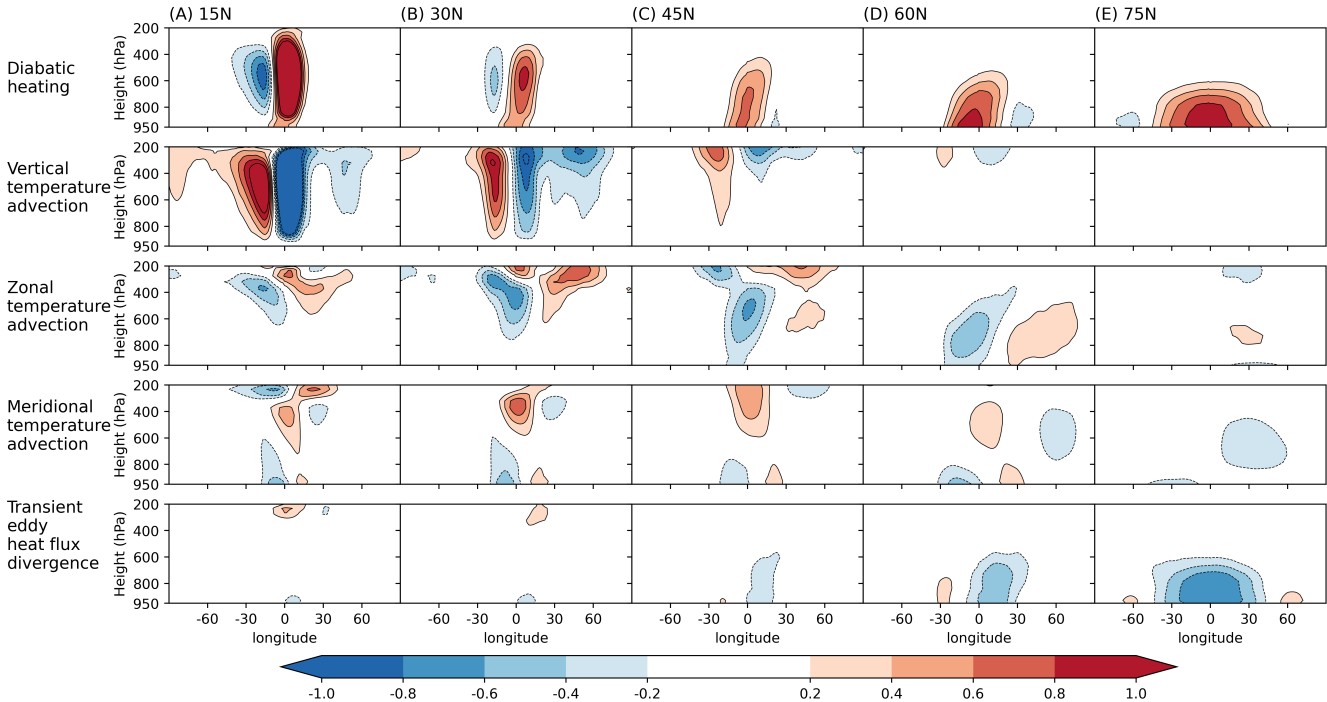

**Figure 6.** The diabatic heating and heat transport terms ($\mathrm{Kday^{-1}}$) in the **(A)** 15°N, **(B)** 30°N, **(C)** 45°N, **(D)** 60°N and **(E)** 75°N heating experiments. The first row shows the diabatic heating; the second row shows the time-mean vertical potential temperature advection; the third row shows the time-mean zonal temperature advection; the fourth row shows the time-mean meridional temperature advection; the fifth row shows the divergence of transient eddy heat fluxes. The longitude-height section shown is a meridional average over ±30° of latitude from the central latitude of the heating perturbations. All transport terms (second to fifth rows) are multiplied by -1 (i.e., moved to the right-hand side of Equation 2) to highlight that they act to balance the diabatic heating.

and latitude from the heating maximum) and the full troposphere (integrating from the surface to 200hPa), with positive ratios indicating that the term acts to offset the perturbation. Moving from low to high latitudes, the time-mean vertical potential temperature advection becomes less important (blue bars; from 117% to -13%) while transient eddy heat flux divergence become more important (green bars; from 5% to 86%) in balancing the heating. The time-mean horizontal advection (red bars) appears to play a minimal role due to large cancellations between the zonal (yellow bars) and meridional (orange bars) components. Furthermore, in the whole-troposphere picture, the meridional advection is mostly negative due to its warming effect in the free troposphere (Figure 6, fourth row). Considering only the lower troposphere (surface to 800hPa), and the western portion (-20° to 0° longitude) of the heating perturbation highlights the importance of the horizontal temperature advection (red bars in Figure 7B), especially for the midlatitude heating (30°N, 45°N and 60°N experiments). In particular, the meridional advection (orange bars in Figure 7B) is important for balancing the 30°N and 45°N heating in the lower troposphere.

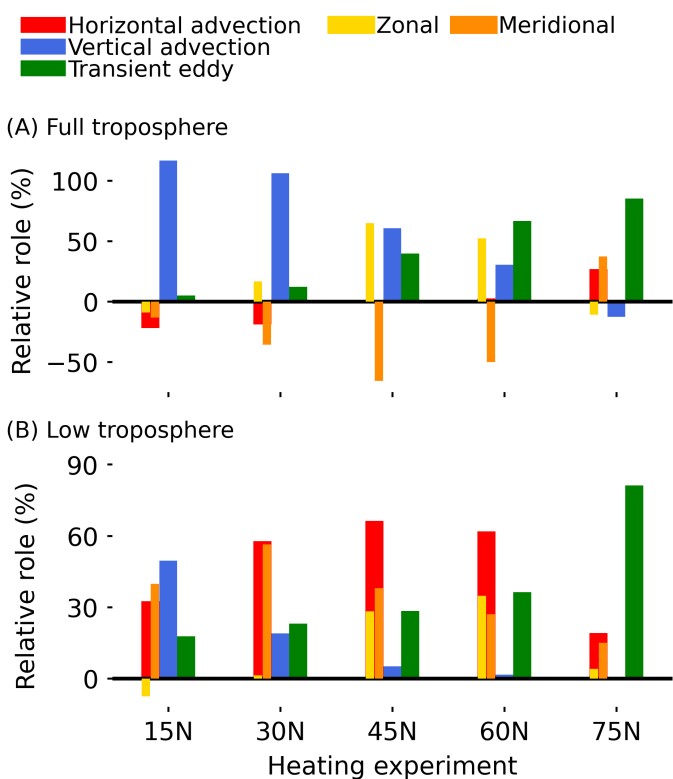

**Figure 7.** Summary of the relative role of the time-mean horizontal temperature advection (red), time-mean vertical potential temperature advection (blue) and transient eddy heat flux divergence (green) in balancing the diabatic heating. The time-mean horizontal temperature advection can be further decomposed into zonal (yellow) and meridional (orange) components. The relative role is the ratio between each heat transport term and the diabatic heating term averaged within **(A)** ±20° of longitude and latitude from the heating maximum and integrated from the surface to 200 hPa and **(B)** -20° to 0° longitude and ±20° of latitude from the heating maximum and integrated from the surface to 800 hPa.

The fact that transient eddies play a dominant role in balancing polar heating might not be surprising given that the background conditions near the pole are not favourable for vertical and horizontal advection. The boundary layer at high latitudes exhibits high static stability and hence vertical motion is inhibited, so balancing via vertical motions is unlikely. The background zonal wind and meridional temperature gradient at high latitudes are weak, so balancing by horizontal advection is also unlikely. Therefore, transient eddies become the only circulation process that can act to diffuse and remove heat from the source region at high latitudes.

How do the transient eddies work to balance the high-latitude heating? Transient eddies are synoptic systems that transport energy polewards. We find that the transient eddy response tends to reduce heat transport from midlatitudes towards the high-latitude heating source rather than fluxing heat further polewards (Figures 8D and 8E). This reduced eddy heat transport is

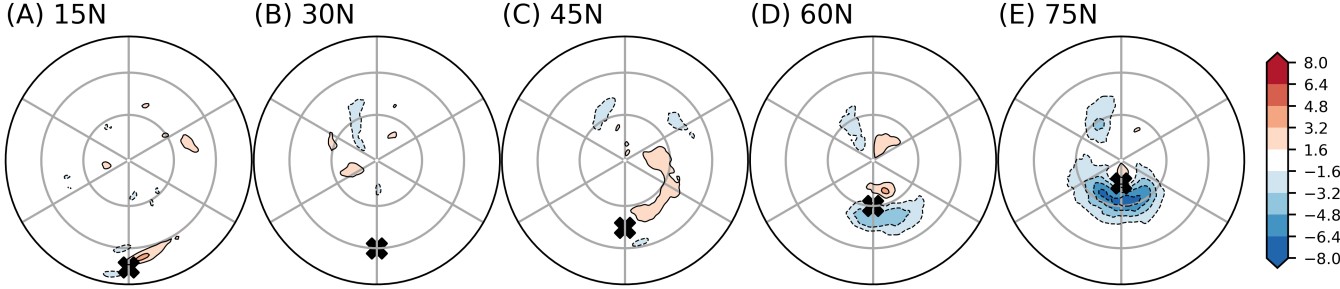

**Figure 8.** Near-surface (990 hPa) meridional transient eddy heat fluxes response ($\overline{v'\theta'}$; Kms$^{-1}$) in the **(A)** 15°N, **(B)** 30°N, **(C)** 45°N, **(D)** 60°N and **(E)** 75°N heating perturbation experiments. The crosses mark the position of the surface heating perturbation. The latitude lines mark 30°N and 60°N. The longitude lines denote 60° intervals, marking 120°W, 60°W, 0°, 60°E, 120°E and 180°.

consistent with reduced baroclinicity equatorward of the heating source, as diagnosed by the Eady growth rate (Figure S6) and the weakened meridional temperature gradient due to amplified polar warming (Figures 2D and 2E).

### 3.4  Contrasting dynamical and radiative adjustment processes

Results to this point show the combinations of circulation processes that are responsible for the dynamical adjustment to heating perturbations at different latitudes. Vertical motion plays the dominant role in balancing heating at low latitudes, while divergence of transient eddy heat flux plays the dominant role at high latitudes. However, these analyses do not speak to why diabatic heating (Figure 6; top row) and the temperature response (Figures 2 and 3) differ as the heating source is moved from low to high latitudes. To investigate this, we turn to radiative cooling, which is hypothesised to be important in the adjustment

of polar regions to energy input (Kim et al., 2021; Woollings et al., 2023; Miyawaki et al., 2023). In this case, a heating perturbation is balanced by enhanced longwave emission due to an increase in atmospheric temperature. Assessing the relative importance of dynamical processes compared to radiative and turbulent fluxes requires us to move from the thermodynamic equation (see section 2.3) to a moist static energy budget of the atmospheric column above the heating source (see section 2.4).

Energy input to the atmospheric column from the Q-flux heating perturbation is accomplished via anomalous upward fluxes

of sensible (Figure 9, red) and latent (orange) heat as well as net longwave radiation (upwards minus downwards; yellow). While the sum of these surface fluxes inputs the same total amount of anomalous energy into the atmosphere across the perturbation experiments (around 40Wm$^{-2}$), the partitioning depends on the latitude of the heating. Latent heat fluxes contribute the most in all experiments, ranging from 93% of the total input for the 15°N heating source to 52% for the 75°N heating source. The decrease in latent heat input is compensated by an increase in sensible heat fluxes and net longwave radiation.

Note that shortwave radiation is not shown because it remains unchanged from the control to perturbation experiments due to the idealised model setup (i.e., no changes in clouds, ice or albedo).

At equilibrium, the anomalous energy entering the atmospheric column from the surface is balanced by anomalous radiative cooling at the top of the atmosphere (outgoing longwave radiation; dark blue) and horizontal energy transport out of the column by the circulation (i.e., the advection and transient terms discussed in previous subsection, treated here as a residual, light blue).

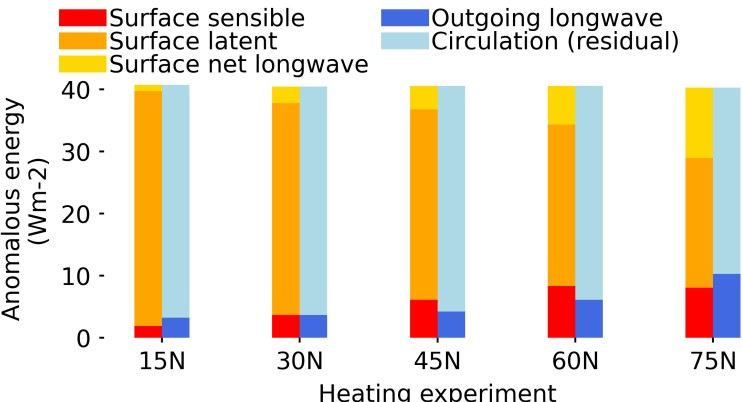

**Figure 9.** Anomalous energy (Wm$^{-2}$) entering (warm colours, left bars) and leaving (cold colours, right bars) the atmospheric column above the heating perturbations compared to the control experiment, averaged over the domain of the heating perturbation. Energy entering the atmospheric column from the surface includes the net longwave radiation (yellow), surface latent (orange) and sensible (red) heat fluxes. Energy leaving the atmospheric column includes the outgoing longwave radiation (dark blue) and the horizontal transport of moist static energy (light blue).

The horizontal energy transport by the circulation plays a more important role than radiative cooling in removing the excess energy locally for all experiments. However, the circulation contribution becomes weaker as the heating is moved further north, and hence, a larger part of the excess energy must be balanced by longwave emission from the top of the atmosphere. An increased reliance on radiative cooling in the balance suggests an amplified temperature response to heating at the poles compared to the tropics (Figures 2 and 3) according to the Stefan-Boltzmann law (i.e., longwave emission is proportional to

$T_e^4$, where $T_e$ is the emission temperature). Such a non-linear relationship further indicates that a larger temperature change is required to emit the same amount of longwave radiation in the colder polar regions than in the tropics (see Henry and Merlis, 2019), which is seen from our experiments as well. For example, the 75°N (15°N) heating experiment shows a 5.5K (0.8K) averaged surface warming with 10.3Wm$^{-2}$ (3.2Wm$^{-2}$) increase of outgoing longwave radiation over the heating source, indicating an additional 1.9Wm$^{-2}$ (4Wm$^{-2}$) longwave emission per degree surface temperature increase. This is consistent with

a less negative Planck feedback that leads to Arctic amplification of warming under globally uniform heating of well-mixed greenhouse gas forcing (Henry and Merlis, 2019).

     A few additional considerations bear mention in explaining the structure of the response to high-latitude heating compared to low-latitude heating. At lower latitudes, vertical advection efficiently moves heat away from the surface and maximises warming in the upper troposphere (Figure 3A). This increases the efficiency of radiative cooling to space, providing a negative

feedback to the surface warming. Conversely, at high latitudes, the lack of vertical advection in moving heat away from the surface implies a bottom-heavy temperature profile (Figure 3E). This steepens the lapse rate and reduces the efficiency of radiative cooling to space, producing a positive feedback that requires more surface warming to restore energy balanc (Bintanja

et al., 2011; Graversen et al., 2014; Pithan and Mauritsen, 2014; Feldl et al., 2020; Boeke et al., 2021). Overall, the dynamical and radiative adjustments contribute to the amplified surface warming for the high latitude heating experiment.

## 4   Discussion and concluding remarks

In this study, we build on the foundational work of Hoskins and Karoly (1981) exploring the atmospheric circulation response to low and mid latitude surface heating. We revisit their study using a moist gray radiation model, and expand the focus to surface heating perturbations at all latitudes from the tropics to the poles. Heating at low to mid latitudes is offset mainly by time-mean vertical and horizontal potential temperature advection, respectively, consistent with Hoskins and Karoly (1981). Additionally, we explore the response to heating at high latitudes, and find the dominant contribution to balancing the heating perturbation is by transient eddies rather than the time-mean circulation. Overall, however, the circulation (time-mean plus transient) response at high latitudes is less efficient at removing near-surface heat compared to the circulation response at lower latitudes. As a result, a greater contribution from radiative cooling is required at high latitudes, leading to an amplified surface and near-surface temperature response. Thus, our idealised modelling results are relevant to understanding the link between surface heating arising from Arctic sea ice loss (analogous to the Q-flux perturbation at 75°N in our idealized setup) and the Arctic amplification of surface warming seen in observations and comprehensive climate models.

Why is the circulation response less efficient in balancing imposed heating at higher latitudes compared to lower latitudes? We hypothesise that the circulation response is more tightly linked to the surface temperature response for the high latitude case. Specifically, the reduced meridional eddy heat transport (Figure 8) is set up by the amplified surface warming via a reduction of baroclinicity (Figure S6). Such a strong dependence limits how much the reduced eddy activity can cool the perturbation region. At lower latitudes, time-mean circulation responses (vertical and horizontal advection) that act to balance the lower-latitude heating are less dependent on the temperature response.

However, future studies should test whether these results are sensitive to the addition of processes that are missing in our idealised setup, including clouds (Kay and Gettelman, 2009; Huang et al., 2021), radiative effect (Tan et al., 2019; Jucker and Gerber, 2017), as well as water vapour and ice-albedo feedbacks (e.g., Beer and Eisenman, 2022; Chung and Feldl, 2024; Feldl and Merlis, 2023). Future studies could test the role of these additional processes using the ISCA modelling framework (Vallis et al., 2018), which allows one to increase the model's complexity step-by-step by adding sea ice, clouds, topography, and a realistic radiative scheme, etc.

Finally, we note that the dynamical and radiative adjustments arising from the perturbations simultaneously shape the temperature responses. As such, our experimental design cannot fully disentangle the cause of the amplified surface temperature response in the "Arctic" heating experiments. A carefully designed experiment with abruptly switched-on surface heating with large ensemble members (similar to the setup in Deser et al., 2007; Hell et al., 2020) but looking at the day-to-day temporal evolution of the dynamic and radiative responses could be helpful in determining the causal relations better than looking at the equilibrium responses as done here. Additionally, incrementally adding dynamic and radiative components into an idealised model might be useful to isolate their individual roles (e.g., Feldl and Merlis, 2021).

*Code availability.* The code to construct all figures will be available online if this manuscript is accepted.

*Data availability.* All data in this study are available upon request.

*Author contributions.* PYFS performed the model simulations, conducted the analysis, prepared the figures, and wrote the paper with contributions from all co-authors. PYFS, CL and SPS conceived of the original idea. CL, SPS, EDS and MT provided guidance on the interpretation
of results.

*Competing interests.* Camille Li is a member of the editorial board of the journal.

*Acknowledgements.* This work was supported by Research Council of Norway projects 255027 (DynAMiTe), 276730 (Nansen Legacy), 295046 (KeyCLIM) and the EMULATE project funded through the Bjerknes Centre for Climate Research by the Norwegian Department of Education. Funding for M. Ting was provided by National Science Foundation grant - AGS 1934358. We acknowledge the ISCA mod-
elling group from the University of Exeter for providing the model framework and support from Norwegian research infrastructure services (NRIS) sigma2 for computing resources. We thank Michael Previdi, Paul Kushner, Tim Woollings, Tyler Janoski and Yutian Wu for useful discussions.

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
