# Peer review of "Supporting Information for "Circulation responses to surface heating and implications for polar amplification""

_EGUsphere, 2023_

## Author Comment (AC1)

**Final author comments - Circulation responses to surface heating and implications for polar amplification (egusphere-2023-3066)**

Peter Yu Feng Siew, Camille Li, Stefan Pieter Sobolowski, Etienne Dunn-Sigouin, and Mingfang Ting

We would like to thank three reviewers for their thoughtful feedback on our manuscript. There were a few concerns that were shared by more than one reviewer and that required special attention. The chief concerns were about (1) the robustness of the results compared to a control simulation that has more realistic transient eddy kinetic energy, (2) how the results might change using more complex models, and (3) clarifying the connection between the thermodynamic equation and the column-integrated moist static energy budget in our analysis. These are important concerns. Point-by-point responses to the comments appear below. Reviewers' comments are in blue, our replies are in black. We assign a reference number to each response so that we can refer back to previous responses where relevant. All figures in this response letter are assigned with the letter 'R' (for example, Figure R1).

RC1: 'Comment on egusphere-2023-3066', Osamu Miyawaki, 16 Jan 2024
General comments:

This paper investigates the circulation, temperature, and atmospheric energy balance responses to surface heating imposed at various latitudes in a moist, gray radiation, clear-sky aquaplanet model. The authors show the circulation responses to surface heating are latitude dependent. The circulation response to low latitude surface heating is characterized by time-mean vertical and horizontal temperature advection whereas the circulation response to high latitude heating is characterized by transient eddy heat flux divergence. They show the composition of the atmospheric energy balance response to surface heating is also latitude dependent, where atmospheric radiative cooling is increasingly important for balancing surface heating with increasing latitude due to the decreasing efficiency of circulation heat export and the temperature-dependence of the Planck feedback.

The analysis and findings presented are interesting and a useful addition to the literature on the circulation response to idealized forcings. Idealized model experiments are valuable for their interpretability but it is important to make clear if the understanding imparted therefrom are applicable to the real world or not. Specifically, I would like to see the authors check the robustness of their results in a control climate that more accurately represents that of the modern climate. I also have other minor comments below that I would like to see addressed prior to publication.

**1. Thank you for the constructive comments. We agree that idealised model experiments are useful to ascertain physical processes. However, it is still important to clarify whether the findings are applicable to more realistic models and the real world. Please see response #3 that aims to address your concern about the robustness of results in a control setup that better represents the modern climate. We also include relevant discussion on how future studies could test the robustness of the results across the model hierarchy (see response #13).**

Specific comments:

Line 4: I suggest specifying "time-mean" or "stationary" circulation responses here to make more clear the contrast with the subsequent discussion on transient eddy heat flux divergence.

**2. In the revised manuscript, we will specify "time-mean" circulation responses, in order to highlight that our results are based on circulation responses under a quasi-steady state.**

Fig. S1A: This approximately factor of 2 discrepancy in the transient eddy kinetic energy climatology between the idealized experiment and reanalysis is quite large. Since the results include the transient eddy heat flux response to surface heating perturbations, I think it is important to check if the results are robust in a model setup that more accurately captures the observed transient eddy kinetic energy climatology of the modern climate. Otherwise, the presented results may only be relevant to that of an equable climate with no sea ice instead of Earth's modern climate. I suspect this discrepancy is likely due to the significantly weaker baroclinicity of a no sea-ice aquaplanet set up. If so, the discrepancy could be reduced by modifying the control climate Q-flux profile such that the resulting baroclinicity resembles that of modern Earth. For example, Miyawaki et al. (2023, Environ. Res.: Climate) introduced a simple way to include the thermodynamic effect of sea ice in the form of a climatological Q flux. Since the resulting meridional temperature profile, even in an aquaplanet with no sea ice model, has been shown to capture that of a climate that has sea ice, I suspect imposing such a Q flux profile will help reduce the discrepancy in transient eddy kinetic energy between the idealized experiment and the reanalysis as well.

**3. We realise that the eddy kinetic energy in the middle troposphere of our control simulation is around two times weaker than observations (Figure S1A). To ascertain whether our main findings are sensitive to this bias, the ideal solution is to rerun all experiments (both control and perturbation experiments) using the Q-flux in Miyawaki et al. 2023 as suggested by the reviewer. However, we do not have extra computational resources to rerun all the simulations. We also still see value in setting up the control simulation following the TRACMIP protocol, which has been widely used and documented (see response #14). Therefore, we suggest an alternative approach that we hope goes some way to addressing these concerns. To support that the main finding is not sensitive to the weak eddy kinetic energy in the control simulation, we break down our analysis into winter and summer seasons (our simulations are run with seasonal cycles but with time-independent Q-flux). Figure R1 shows that eddy kinetic energy in both winter (JFMA) and summer (JASO) of the control is weaker than the observations, but the difference between the two seasons is comparable to the difference between the control and observations (also around two times). Figure R2 shows that the main finding (i.e., transient eddy heat flux divergence is the dominant circulation process balancing high-latitude heating) is consistent in the two seasons, suggesting that the main finding is not sensitive to the bias in eddy kinetic energy in the control simulation. In the revised manuscript, we will mention the main result is not sensitive to this seasonality result, and highlight that we can produce an eddy kinetic energy field more similar to reanalysis in the control by using the adjusted Q-flux from Miyawaki et al. 2023.**

[Figure]

**Figure. R1:** As in Figure S1A, but showing the transient eddy kinetic energy ($m^2s^{-2}$) in summer (left column) and winter (right column) for control (upper row) and NCEP Reanalysis (lower row).

[Figure]

**Figure R2:** As in Figure 7, but showing the relative role of each circulation term in balancing the heating perturbations in summer (upper row) and winter (lower row).

Fig. S1 right column: It's surprising to see the lack of a near-surface inversion over the high latitudes in the reanalysis. I was expecting this to be one of the discrepancies between the temperature profiles of an idealized model with no sea ice and reanalysis. How is subsurface

data treated for reanalysis data? The fact that 1000 hPa data are shown over Antarctica makes me wonder if subsurface data are not masked.

**4. Yes, the subsurface data was not masked in the reanalysis data. Now we do this. The near-surface temperature inversion in the Arctic is still not obvious in the annual mean, but is obvious in winter (JFMA, see the following Figure R3). In the revised manuscript we will revise this figure by masking the subsurface data.**

[Figure]

**Figure R3:** As in Figure S1B, but showing the annual (left) and winter (right) climatological temperature profile (in height-latitude sections) for the NCEP R2 reanalysis data.

Fig. 1: Why is the outermost contour of each Q-flux profile not smooth? This seems unexpected from equation 1.

**5. The Q-flux profiles were created at a rather low resolution to fit the model resolution (T85, around 1.4 latitude x 1.4 longitude grid). We now smooth the lines when making the plot (see Figure R4), and will revise this figure in the manuscript.**

[Figure]

**Figure R4:** As in Figure 1, but making the lines smoother.

Fig. 9 and Section 3.4: I think it's important that these results are accompanied by an additional methods subsection on the column-integrated atmospheric energy budget. The terms in Fig. 9 are not defined in the current manuscript. This new method subsection should make clear that the circulation (residual) term differs from the temperature advection terms in equation 2 because of the inclusion of other advected energy terms such as latent and geopotential energy in Fig. 9.

**6. Thank you for the suggestion. We will include an equation that shows the column-integrated moist static energy budget of the atmosphere. The circulation term ($\nabla \bar{u} \cdot E$), which is treated as the residual, represents the transport of moist static energy. We will include a new subsection in the Methods to introduce the moist static energy budget and what the terms represent. We will also highlight that the circulation term here differs from the temperature advection terms in the thermodynamic equation.**

$$F_{surface} = F_{Top} - \nabla \bar{u} \cdot E$$

Equation R1: $F_{surface}$ describes the net surface turbulent (sensible and latent) and radiative fluxes. $F_{top}$ describes the top-of-the-atmosphere radiative fluxes. $\bar{u}$ describes the vertical integral of zonal and meridional winds; E describes the vertical integral of moist static energy. $\bar{u} \nabla \cdot E$ indicates the divergence of the vertical integral of moist static energy.

Fig. 9 and Section 3.4: Related to above, it would be helpful to have a discussion somewhere (perhaps in the aforementioned new method subsection) on why this section considers the moist static energy budget (where surface latent heat flux is a diabatic term) as opposed to the dry static energy budget (where column-integrated condensation, or precipitation is an adiabatic term). Considering the results up to this point are based on the dry static energy budget, why not keep things simple by sticking to the dry static energy budget for Fig. 9 and Section 3.4?

**7. We calculate all circulation terms in the thermodynamic equation to quantify their roles in balancing the diabatic heating, following Hoskin and Karoly (1981). As mentioned in the methods section (lines 97-100), the diabatic heating term calculated directly from source terms matches well with the diabatic heating term calculated as a residual of the transport terms. The thermodynamic equation is particularly helpful in looking at the balance level by level, as in Figure 6 of the manuscript, or contrasting the balance between different layers of the atmosphere, as in Figure 7 of the manuscript. In Figure 9 and section 3.4, our goal is to quantify/compare the relative importance of radiative and circulation processes for the entire atmospheric column, and for this purpose, it is natural to use the moist static energy budget (see response #6). Furthermore, using moist energy budget and treating the circulation term as a residual is a common practice in previous studies that quantified the role of atmospheric energy transport in contributing to Arctic Amplification (e.g., Pithan and Mauritsen 2014; Kim et al. 2021), making it more straightforward to put our results in context. As mentioned in response #6, we will include a new paragraph in the Method to introduce the moist static energy budget, and will modify the writing to make a better transition from subsections 3.3 (using thermodynamic equation) to 3.4 (using moist static energy budget).**

Line 217-220: This sentence is confusing to me. Isn't the amplified high-latitude temperature response plausible due to an increased reliance on radiative cooling in the high latitudes in addition to the temperature dependence of the Planck feedback? They seem like two separate contributions/mechanisms to me but this sentence reads as if they are somehow related. If so, can this link be further elaborated?

**8. We would argue that the two are related contributions. According to the Stefan-Boltzmann law, the longwave emission (radiative cooling) is proportional to emission**

temperature to the power of four. This indicates that a stronger reliance on longwave cooling results in a larger surface temperature response. The nonlinear relationship in the law further indicates an amplified surface temperature response at higher latitudes for a given increase in longwave emission because of the colder background temperature in the pole (which is a result of the Planck feedback). Reviewer 3 gives similar suggestions to improve the connection between these two sentences (see response #27). We will modify the writing to make the connection clear.

Line 244-248: I think it's important to specify here that you are referring to the surface temperature response. The results show the circulation has an important impact on the vertical structure of the temperature response in both the low and high latitudes.

**9. We agree. The dependence between the reduction of meridional eddy heat transport and amplified temperature responses mostly happens on the surface. In the revised manuscript we will specify that we are referring to the surface temperature response.**

Line 249-255: Given the highly idealized model setup of this study, I think it would be useful to specify how future studies could further investigate the robustness of the results across a hierarchy of complexity, such as the role of radiation spectroscopy (i.e., non-gray radiation) and clouds, in particular.

**10. We agree. Reviewers 2 and 3 raise similar concerns related to how the results might be sensitive to the missing components (full radiation, water vapour feedback, cloud, etc.) in our idealised setup. Please see responses #13 and #31 to these concerns. In short, we will introduce a new paragraph in the Discussion to briefly discuss how the results might be affected by processes not included in the model, and outline a plan encouraging future studies to test the results across model hierarchy.**

Technical comments:

Line 213, 215, and Fig. 9 caption: I believe the term "lateral" energy transport is used here to mean "horizontal" energy transport, which is the term that has been used throughout the manuscript up to this point. For consistency I suggest rephrasing lateral to horizontal.

**11. Yes, they mean "horizontal" energy transport. We will make changes in the revised manuscript.**

RC2: 'Comment on egusphere-2023-3066', Anonymous Referee #2, 01 Feb 2024
Summary

This study uses an idealized moist GCM to examine the qualitative dependence of the atmospheric response to a localized large-scale surface heat source on the latitude of the heat source. The experimental setup is an idealized aqua-planet setup, without land, ice and clouds and with a gray-radiation parameterization. The equilibrated response is analyzed in terms of temperature, circulation and energy balance. The main findings, which are generally consistent with previous studies, show that the response in the tropics is mainly through convective processes, which transfer the heat upward and lead to energy flux out of the air column, whereas at high latitudes the response shows a strong heating of the surface and

lower troposphere, which leads to increased outgoing longwave radiation. The authors suggest a link between these results and the phenomenon of polar amplification.

The manuscript is well-written, and the rational of the study is clear. I find the method adequate for addressing the research question. However, I find the discussion and concluding remarks a bit too vague and too short. I think this manuscript is fit for publication in WCD, in terms of the scientific scope. I suggest a few specific (rather minor) revisions prior to publication, as elaborated below.

**12. Thank you for your comments, which allow us to reflect on the weaknesses in the presentation of our study. Please see our following point-to-point responses that aim to extend and clarify our discussion and concluding remarks.**

Major comments

Description of the model setup (subsection 2.1): Some details of the model setup and some discussion of those details are missing. (a) Does the model include a representation of clouds and their radiative effect? I suppose it doesn't, but there is no discussion about the choice of using a model without clouds to study the energetic response to surface heating. Would you except to find significantly different results if clouds were included? It could affect the level from which longwave radiation is emitted to space, and thus affect the relation between surface temperature and OLR (lines 218-219). (b) It is mentioned that the model doesn't include sea ice, but again – it is not mentioned whether the implications of the results for Arctic warming are affected by the absence of sea ice in the model. (c) What is the motivation for using a gray radiation model rather than a full radiation model, that gives a more realistic circulation (see Jucker and Gerber, 2017 and Tan et al. 2019)?

References

Jucker, M. and E. Gerber, 2017: Untangling the annual cycle of the tropical tropopause layer with an idealized moist model. Journal of Climate, 30 (18), 7339-7358.

Tan, Z., O. Lachmy and T. A. Shaw, 2019: The sensitivity of the jet stream response to climate change to radiative assumptions. J. Adv. Model. Earth Syst., 11 (4), 934-956.

**13. The model has no cloud, no land, and only includes a gray radiation scheme. We fully acknowledge that the presence of clouds, sea ice, and full radiation are important to understand the temperature and circulation responses to heating perturbations and in the real world. Nevertheless, the rationale of this study is to build on the study of Hoskins and Karoly 1981, which investigated circulation responses to low and midlatitude heating perturbations using a very simplified model framework (dry dynamical core). Here we extend their study using a more complex model by adding moist and simplified radiation processes. The results of low- and mid- latitude heating from the simplified dry model (used in Hoskins and Karoly 1981) and from our moist, gray radiation model model are consistent, suggesting that the basic dynamic response is captured in a dry model. But the added complexity (although not as complex as it could be) becomes useful in understanding the response to high latitude heating, which was not a focus of Hoskins and Karoly 1981, allowing us to parse the radiative and moist processes in addition to the dry dynamics. Our intention here is not to**

run a realistic GCM, but to use a simplified framework to improve our understanding of a complex system. We leave it to future studies to test how increasing the complexity of the model might change the results. In the revised manuscript, we will add a paragraph in the Discussion section to recognize the importance of these missing model components for understanding atmospheric responses with relevant discussion. We will also encourage future studies to test the results across the model hierarchy. The ISCA modelling framework we are using can facilitate this, as it allows one to increase the complexity step-by-step by adding sea ice, land, a more realistic radiative scheme, etc. (see Vallis et al. 2018).

(d) Why do you choose to use the TRACMIP protocol with diurnal and seasonal cycles? Eventually, only the climatological annual mean response is considered, so why do you choose to include the full cycles? (e) I suppose there is no land in these experiments, but it is confusing that the TRACMIP initials include the word "continent". Is there land in the model or not?

**14. Our model setup is an Aquaplanet (i.e., no land). Although TRACMIP focuses on the tropical rain belt over the continent, previous studies have used it for Aquaplanet. The TRACMIP paper (Voigt et al. 2016) included two Aquaplanet models, one of which (they called it the "CALTECH model") is the same model we use here. The TRACMIP protocol has also been used for Aquaplaent setups in more complex models - e.g., NCAR CESM2 (Dunn-Sigouin et al. 2021). The setup in TRACMAP has several advantages. For example, it includes the seasonal cycle of solar radiation, which allows us to explore whether our results are sensitive to biases in the background conditions (see our response #3 to reviewer 1).**

Equation 2: The time tendency term is omitted. It would be good to at least mention that this is an approximate equation, assuming a steady state. It is mentioned a few lines below that Q is calculated as a residual, and that it's vertical integral is very close to the vertical integral of the diabatic heating, calculated as the sum of all the source terms. It is argued that "This confirms that the residual method provides a good estimate of the diabatic heating", but it is not mentioned that this depends on the assumption that the system is in steady state, and that the variables are averaged over a long enough period so that the tendency term is negligible. Later, in figure 6, the terms in this equation are shown as a function of longitude and height. Is the tendency term negligible also locally or only when considering the vertical integral?

**15. The tendency term is expected to be small at all levels over a "long enough" averaging period. To confirm this, we select two different levels (990 and 500 hPa) over the heating sources in each heating experiment and see how the temperature changes over the 30-year integration period (Figure R5). Results show that the temperature is very steady over the whole period, confirming that the temperature tendency is negligible in the long-term average. In the revised manuscript, we will further clarify the assumption of a steady state so that the time tendency term can be omitted.**

[Figure]

**Figure R5:** Air temperature at 500 (orange) and 950 (red) hPa over the heating centres from 15N to 75N heating perturbation experiments.

The temperature response is shown at the surface as a function of longitude and latitude (figure 2) and as a longitude-height cross-section (figure 3), but the latitude-height profile is not shown. It would give a more complete picture to see the latitudinal distribution of the temperature response, not just at the surface, but also throughout the troposphere. The energy budget (figure 9) implies that in the tropical heating case, the energy is transported away from the source region. Does this energy transport heat the atmosphere at higher latitudes? A latitude-height cross-section of the temperature response would show that.

**16. Here we provide the latitude-height cross-section of temperature (Figure R6) and meridional heat transport (Figure R7) responses. We agree that these figures give us an additional perspective of how heat is transported away from the source region. For the 15N heating experiment, the strong positive temperature response in the upper troposphere and the remote temperature responses suggest that meridional transport removes heat away from the atmospheric column in the upper level. This can be further confirmed in Figure R7, showing that there is a strong upper-level divergence at 15N that brings heat away from the atmospheric column. For the 75N heating experiment, the reduction of meridional heat transport is consistent with the reduction of meridional transient eddy heat flux (Figure 8 in the manuscript) that acts to balance the heating perturbation. In the revised manuscript we will include these figures in the supplementary with relevant discussion.**

[Figure]

**Figure R6:** Air temperature response (K) in the (A) 15°N, (B) 30°N, (C) 45°N, (D) 60°N and (E) 75°N heating perturbation experiments. The latitude-height section is shown as the zonal average over the longitudes of the Q-flux perturbations.

[Figure]

**Figure R7:** Meridional heat transport (vT, Km/s) in the (A) 15°N, (B) 30°N, (C) 45°N, (D) 60°N and (E) 75°N heating perturbation experiments. The latitude-height section is shown as the zonal average over the longitudes of the Q-flux perturbations.

Subsection 3.4 shows the vertically-integrated atmospheric energy budget (figure 9). It would help the reader if the relevant equation would be written explicitly. The residual is said to be equal to advection of energy by the time-mean circulation plus energy transport by transient eddies (line 214). If the full energy budget equation would be written down, it would help to see what exactly these terms are. It is said that these terms are discussed in the previous subsection, but there it was part of the potential temperature equation, which is not the same as the vertically-integrated atmospheric energy budget.

**17. Reviewer 1 raises the same concern, and please see our responses #6 and #7 for details. In short, we will add a new subsection in the Methods to describe the moist static energy budget. We will also highlight that the potential temperature advection is part of, but not equal to the column-integrated moist energy transport (treated as the residual).**

Discussion and concluding remarks (section 6): This section contains statements that are not clear, and that their connection to the results is not clear (lines 240-242, 246-248). How are these results relevant to the connection between Arctic sea ice loss and Arctic amplification, if there is no sea ice in the model?

**18. The reduction of Arctic sea ice has been hypothesised to influence atmospheric circulation and temperature. The fundamental step to triggering the pathway is that Arctic sea**

ice reduction exposes the relatively warm ocean and adds energy into the atmosphere via surface energy fluxes. Our perturbation experiments impose surface heating via Q-flux, which also inputs energy into the atmosphere via surface fluxes (reddish bars in Figure 9). Therefore, our idealised setup is analogous to how sea ice reduction influences the atmosphere in more complex models and the real world. In the revised manuscript we will make this connection clearer.

Why does the reduced baroclinicity in the case of the high-latitude heating perturbation limit how much the reduced eddy activity can cool the perturbation region? Why are vertical and horizontal circulation responses expected to be less dependent on the temperature response at lower latitudes?

**19. Here we aim to raise a physical explanation of why transient eddies are less efficient than the time-mean circulation in balancing the heating perturbations. Here we propose that the time-mean circulation responses (i.e., vertical and horizontal temperature advection) that act to balance the lower latitude heating are not dependent on the temperature response itself. In contrast, the transient responses that act to balance the high-latitude heating are dependent on the temperature response via the reduction of baroclinicity. We will make this point more clear in the revised manuscript.**

Additionally, I would expect to find here some discussion about the limitations of the relevance of these results to the actual atmosphere, due to the absence of clouds and sea ice in the model.

**20. Please see response #13. In the revised manuscript we will add relevant discussion.**

Further, it is not quite clear what part of the results is new, and what part is consistent (or non-consistent) with results of previous studies.

**21. Our study builds on Hoskins and Karoly (1981) which investigated the circulation responses to low- and mid-latitude heating perturbations in a dry model. Our findings are consistent with what they found even though the models used are quite different: the low latitude heating is balanced by vertical heat advection while the midlatitude heating is balanced by horizontal heat advection. This suggests that dry dynamics dominate the response in these perturbation experiments. The new part is that we extend the heating perturbation to high latitudes (our 60N and 75N heating experiments). We find that transient eddy heat flux divergence is the main circulation process balancing the heating. We further quantify the role of radiative cooling, and find that there is a stronger reliance on radiative cooling as an additional process for balancing the heating at higher latitudes (the role of radiative cooling was hypothesised in previous studies but not explicitly tested). In the revised manuscript, we will further modify the first paragraph in the Discussion to make clear which results are new.**

Minor comments

Figure S4: I think this figure would be more appropriate to include in the main paper, rather than the supplementary material. It would also require adding some text to explain what it

means. This is just a suggestion. But if not, perhaps it would be better to remove it, because showing the vorticity response without an explanation is not very informative.

**22. Although the remote upper-level Rossby wave response is not the focus of the study, some readers would be interested to see that, especially when polar surface heating such as sea ice reduction is hypothesised to influence midlatitude circulation via equatorward Rossby wave propagation (e.g., Honda et al. 2009). Showing the vorticity response is useful in the sense that they can be directly compared with Hoskins and Karoly (1981) which also showed the vorticity responses as a proxy for Rossby wave propagation (see Figure R8 below). The upper-level Rossby wave responses share some similarities between their and our results, even though the models and how the heating is imposed in the two studies are quite different. In the revised manuscript, we tend to keep this figure in the supplementary (because the remote response is not the focus of this study), but will add a short sentence highlighting the similarity of the responses compared to Hoskins and Karoly 1981.**

[Figure]

**Figure R8:** Upper-level vorticity response for 15N (left column) and 45N (right column) heating experiments from Hoskins and Karoly 1981 (upper row - from their Figures 3B and 4B) and from our current study (lower row - from our Figure S4).

Lines 140-141: "Overall, mean meridional and vertical advections do not appear to play important roles in balancing high-latitude, near-surface heating perturbations". At this point, it is still not shown, it is shown in the following sections.

**23. We agree. We will mention that this will be further confirmed in the next section.**

In all the places where "vertical temperature advection" is mentioned (e.g., Figure 6 – panel title and caption and lines 148, 237), it should be called "vertical potential temperature advection". There is a great difference between vertical temperature advection and vertical potential temperature advection, as the latter includes the effect of heating/cooling by contraction/expansion.

**24. We agree. In the revised manuscript we will change to use "vertical potential temperature advection" in the places you mentioned.**

Line 155: the horizontal temperature advection has a similar role in the 15N and 30N simulations (Figure 6A,B), why do you say that it starts to play a role in the 30N simulation?

**25. We agree. We will correct this sentence and mention that for both 15N and 30N simulations, the horizontal temperature advection provides some minor roles, especially in the lower troposphere.**

The use of the word "help" (lines 163, 174) indicates that moving excess heat or offsetting the perturbation is a "good" thing. I suggest to replace with "moves excess heat" and "offsets the perturbation" or "acts to offset the perturbation".

**26. We agree. We will make the suggested changes in the revised manuscript.**

Lines 220-221: It would help the reader to explain what "such a relationship" means, i.e., that because of the nonlinearity of the Stephan-Boltzmann relation between longwave radiation and temperature, for a given change in the longwave radiation, the temperature change required to create this change is greater at lower temperatures.

**27. We agree. We will make the relevant change along with the suggestion by reviewer 1 (see responses #8).**

Lines 229-230: It is not clear what you mean here by "lapse rate feedback" in the context of stronger near-surface warming at high latitudes and how it is related to the lack of vertical advection. This requires some explanation.

**28. We agree that the connection is not clear enough. In the revised manuscript, we will further explain how the lack of vertical heat advection at high latitudes leads to a bottom-heavy temperature response, increasing the lapse rate and decreasing the emission to space (positive lapse rate feedback).**

Language / typos

Line 106: "from the upper to lower troposphere" – add "the" before "lower". Line 215: "energy transport by circulation" – add "the" before "circulation".

**29. Thank you and we will fix them in the revised manuscript.**

**RC3: 'Comment on egusphere-2023-3066', Anonymous Referee #3, 14 Feb 2024**
**Title: Circulation responses to surface heating and implications for polar amplification**

Authors: Peter Yu-Feng Siew, Camille Li, Stefan Pieter Sobolowski, Etienne Dunn-Sigouin, and Mingfang Ting

Summary: This study employs a moist, gray radiation model to investigate the atmospheric response to tropospheric heating perturbations at different latitudes. The findings largely align with those presented by Hoskins and Karoly (1981), but additionally shed light on the

significance of transient eddy heat flux divergence in high-latitude scenarios, particularly. The subject matter is both intriguing and significant, enhancing our comprehension of the dynamic impacts of high-latitude thermal forcings. Overall, the manuscript is well-written. I provide a few major and minor comments below for the authors' consideration.

Major comments:

In the initial sections of the manuscript, the authors scrutinized the complete thermodynamic equation and asserted that transient eddy heat flux divergence is the primary mechanism influencing polar circulation response (as depicted in Figure 6). However, in the subsequent portions, where the authors compared dynamical and radiative adjustment processes, the importance of the circulation component decreases as the forcing location shifts to higher latitudes (as illustrated in Figure 9). How could we reconcile these seemingly conflicting results? I guess there may be not necessary of conflict, but just interpretation from different perspective. Could the authors offer additional explanation or engage in further discussion?

**30. Thank you for pointing this out. We would like to highlight that Figures 6 and 9 are not conflicting results. Figure 6 shows that the transient eddy heat flux divergence plays the dominant role in balancing the heating perturbation at higher latitudes. Although radiative cooling becomes gradually more important for the balance when the heating is placed further north (Figure 9), the circulation is always more important than radiative cooling for the balance (i.e., the light blue bars are always longer than the dark blue bars in Figure 9 for all heating experiments). This suggests that circulation is less effective in balancing the heating perturbations at higher latitudes, leading to a stronger reliance on radiative cooling for the balance. In the second paragraph of the Discussion, we further propose why the circulation becomes less effective for the balance at higher latitudes (please also see response #19). In the revised manuscript, we will make changes in the summary (first paragraph in the Discussion) to highlight that the results shown in Figures 6 and 9 do not conflict with each other. Please also see our response #7 related to the transition from Figure 6 (using thermodynamic equation) to Figure 9 (using moist static energy budget).**

In the model utilized for this study, the absence of water vapor feedback is notable due to the imposition of a single optical thickness across the entire long wave frequency band. However, recent studies have delved into the role of water vapor feedback in both attenuating and intensifying polar amplification (e.g., Beer and Eisenman 2022; Chung and Feldl 2023; Feldl and Merlis 2023). I am intrigued by how the incorporation of water vapor feedback might alter the magnitude and spatial distribution of polar amplification and subsequently influence atmospheric circulation responses. Moreover, considering the direct impact of water vapor on atmospheric circulation, could the authors provide further insights into these aspects?

References:

Beer, E. and Eisenman, I., 2022. Revisiting the role of the water vapor and lapse rate feedbacks in the Arctic amplification of climate change. Journal of Climate, 35(10), pp.2975-2988.

Chung, P.C. and Feldl, N., 2023. Sea ice loss, water vapor increases, and their interactions with atmospheric energy transport in driving seasonal polar amplification. Journal of Climate, pp.1-28.

Feldl, N. and Merlis, T.M., 2023. A Semi-Analytical Model for Water Vapor, Temperature, and Surface-Albedo Feedbacks in Comprehensive Climate Models. Geophysical Research Letters, 50(21), p.e2023GL105796.

**31. Thank you for providing the reference papers that highlight the impact of water vapour feedback on amplified Arctic warming. We recognize that water vapour feedback is important for circulation and temperature responses. Reviewers 1 and 2 also raised similar concerns - how the missing but important processes in our idealized setup (i.e., clouds, full-radiation and sea ice) might affect the results. Please see response #13 for details. In short, we will add a paragraph in the Discussion to recognize the importance of the missing processes (e.g., water vapour feedback) with relevant literature suggested by the reviewers. We also encourage future studies to test the results along model hierarchy (i.e., from idealised to more complicated models). We propose that using the ISCA framework is well-suited for this.**

Minor comments:

Lines 8-10: how could tell the cause and effect? Is sea ice loss a cause or effect?
**32. In the real world and fully coupled models, sea ice and atmosphere influence each other: atmospheric warming contributes to sea ice decline; sea ice decline also contributes to surface warming. Nevertheless, our study implicitly treats sea ice loss as a cause: how the surface heating (analogous to sea ice decline) affects the atmosphere.**

Line 70: why the eddies in the upper troposphere appear to be weaker in the model? Is this related to vertical resolution in the model?

**33. As reviewer 1 suggested, it could be related to the absence of sea ice in the model. We agree that the vertical resolution in the model might also affect the transient eddies (Frierson et al. 2006 showed sensitivity of transient eddy kinetic energy to horizontal resolution). We further break down the analysis in winter and summer and our main findings are not sensitive to the weak transient eddy kinetic energy in the control. Please see our response #3 for details.**

Line 84: how do you derive 450TW?

**34. We integrate the Q-flux (Wm$^{-2}$) profile over its areal extent (m$^2$) (Figure 1 or R4). By design, all Q-flux perturbations at various latitudes have the same size and output the same amount of energy (about 450TW).**

Line 149: why the sign of the sum of the heart transport terms (blue color shading) is opposite from that of the diabetic heating (red color shading)? Is there a missing factor of -1? Line 149 and Figure 6: could the authors plot out the sum of the heart transport terms?

**35. Thank you for pointing this out! The sum of the heat transport terms is exactly equal to the diabatic heating (the diabatic heating term is treated as a residual in the thermodynamic**

equation - see Equation 2 in the manuscript). In Figure 6 we deliberately reverse the sign (by multiplying a factor of -1) for all transport terms to highlight that the circulation terms act to balance the diabatic heating. In the revised manuscript we will make this clear in the caption of Figure 6.

Line 155: in the 30N case, the horizontal temperature advection has already presented some signals? And the role of eddies seems somewhat similar in 30N and 15N cases near surface.

**36. We agree. Reviewer 2 raises a similar concern (see response #25). In the revised manuscript, we will highlight that horizontal temperature advection has already made some minor contributions in both 15N and 30N heating experiments, and the role of transient eddies appears to be negligible.**

Line 225: what does the global uniform heating refer to as? Well-mixed GHG forcing?

**37. Yes, here we want to highlight that our heating source is very localised, different from global warming where the heating is more uniform because of the well-mixed greenhouse gas forcing. In the revised manuscript we will change the phase to "global uniform heating under well-mixed GHG forcing".**

References:

Miyawaki, O., T. A. Shaw, and M. F. Jansen. "The emergence of a new wintertime Arctic energy balance regime." Environmental Research: Climate 2.3 (2023): 031003.

Voigt, Aiko, et al. "The tropical rain belts with an annual cycle and a continent model intercomparison project: TRACMIP." Journal of Advances in Modeling Earth Systems 8.4 (2016): 1868-1891.

Vallis, Geoffrey K., et al. "Isca, v1. 0: A framework for the global modelling of the atmospheres of Earth and other planets at varying levels of complexity." Geoscientific Model Development 11.3 (2018): 843-859.

Dunn-Sigouin, E., C. Li and P.J. Kushner, Limited influence of localized tropical sea-surface temperatures on moisture transport into the Arctic, GRL, 48, e2020GL091549, 2021

Hoskins, Brian J., and David J. Karoly. "The steady linear response of a spherical atmosphere to thermal and orographic forcing." Journal of the atmospheric sciences 38.6 (1981): 1179-1196.

Pithan, Felix, and Thorsten Mauritsen. "Arctic amplification dominated by temperature feedbacks in contemporary climate models." Nature geoscience 7.3 (2014): 181-184.

Honda, Meiji, Jun Inoue, and Shozo Yamane. "Influence of low Arctic sea-ice minima on anomalously cold Eurasian winters." Geophysical Research Letters 36.8 (2009).

Kim, Doyeon, et al. "Atmospheric circulation sensitivity to changes in the vertical structure of polar warming." Geophysical Research Letters 48.19 (2021): e2021GL094726.

Frierson, Dargan MW, Isaac M. Held, and Pablo Zurita-Gotor. "A gray-radiation aquaplanet moist GCM. Part I: Static stability and eddy scale." Journal of the atmospheric sciences 63.10 (2006): 2548-2566.

---

## Referee Report (RR1)

Review of "Circulation responses to surface heating and implications for polar amplification"

**General comments:**
The authors have adequately addressed my previous comments. I recommend the manuscript be approved for publication on the condition that they address my follow-up comments below.

**Specific comments: (line numbers refer to the tracked changes version)**
Line 114-115: Here you state u and E are individually vertically integrated quantities and also that div(uE) is the divergence of vertically integrated moist static energy. This implies that div(<E>)=div(), where < > denotes the mass-weighted vertical integral. I don't think this is generally true. I suggest you explicitly show where the vertical integral operator is in the equation and remove the prefix "vertically integrated" when describing u and E.

Line 117: Moist static energy does not include kinetic energy, hence the label "static" energy.

Line 241-243 and 269-272: The way these statements are phrased implies a direction of causality that the weaker circulation response drives the larger radiative cooling response. Do you have evidence to support the direction of causality? If not, I suggest rephrasing to eliminate the implication of causality; e.g., "The weaker circulation contribution in the high latitudes is *associated with* a stronger radiative cooling contribution."

---

## Author Response (AR2)

**Second revision - final author comments - Circulation responses to surface heating and implications for polar amplification (egusphere-2023-3066)**

Peter Yu Feng Siew, Camille Li, Stefan Pieter Sobolowski, Etienne Dunn-Sigouin, and Mingfang Ting

We would like to thank the reviewers again for their feedback. Point-by-point responses to the comments below. Reviewers' comments are in blue, our replies are in black. We highlight the line number changes in the track-changed manuscript in red.

Reviewer 1: Review of "Circulation responses to surface heating and implications for polar amplification"
General comments:

The authors have adequately addressed my previous comments. I recommend the manuscript be approved for publication on the condition that they address my follow-up comments below.

**1. Thank you very much.**

Specific comments: (line numbers refer to the tracked changes version)
Line 114-115: Here you state u and E are individually vertically integrated quantities and also that div(uE) is the divergence of vertically integrated moist static energy. This implies that div(<E>)=div(), where < > denotes the mass-weighted vertical integral. I don't think this is generally true. I suggest you explicitly show where the vertical integral operator is in the equation and remove the prefix "vertically integrated" when describing u and E.

**2. Thank you for raising this mistake. The vertical integral should be outside of the product of u and E (i.e., firstly calculate the product of u and E at all individual levels, and then take the vertical integral). Nevertheless, we have treated this term as the residual so the results will not be changed. We have now modified the presentation of the equation. Please see Equation 3 and lines 113-115 in the revised manuscript with track changes.**

Line 117: Moist static energy does not include kinetic energy, hence the label "static" energy.

**3. This was also a mistake. We removed the kinetic energy within the bracket. See line 117.**

Line 241-243 and 269-272: The way these statements are phrased implies a direction of causality that the weaker circulation response drives the larger radiative cooling response. Do you have evidence to support the direction of causality? If not, I suggest rephrasing to eliminate the implication of causality; e.g., "The weaker circulation contribution in the high latitudes is associated with a stronger radiative cooling contribution.

**4. We do not intend to imply causality between the circulation and radiative processes. We have now modified a few phases to eliminate the implication of causality. Please see line 239 and line 265.**

Reviewer 2: Review of "Circulation responses to surface heating and implications for polar amplification"

The authors have addressed all my comments in a satisfactory way. I find the paper appropriate for publication in WCD, except for a few minor issues elaborated below.

**5. Thank you very much.**

Specific comments
1) I haven't noticed it in the previous round, but I think equation (1) is missing something. Looking at the Q-flux in Thompson and Vallis (2018) I see that they set it to a constant negative value in all the places where the expression in equation (1) is negative. Is that done also here? Because otherwise it will become increasingly negative with the distance from the heating center, and this doesn't seem to be the case.

**6. Thank you for catching this. We set the Q-flux perturbation outside of the paraboloid function (i.e., the negative values) to be zero. We have modified line 84 to reflect this in the revised manuscript (with track changes).**

2) In the text explaining equation (3) it says that u is the vertical integral of the zonal and meridional winds and E is the vertical integral of moist static energy, so that the second term on the RHS of equation (3) is the transport of the vertical integral of moist static energy. This equation is wrong, because there should be a contribution from product of the vertically-varying u and E. The vertical integral of the energy transport should be taken after multiplying u and E. For example, the energy transport by the Hadley cell is mostly due to the correlation between the meridional wind, which changes from equatorward to poleward with height and the MSE which increases with height. Taking the vertical integral before multiplying u and E would result in a much weaker energy transport. I understand that this term was calculated as a residual, so this mistake has no effect on the results, but nonetheless it should to be corrected.

**7. This was a mistake also noted by reviewer #1. Please see response #2. We have modified Equation 3 and the relevant text.**

3) Figure 6: The subtitle of the second row should be "Vertical potential temperature advection" instead of "Vertical temperature advection".

**8. We have now modified the figure. See the new Figure 6.**

4) The last paragraph of section 3.4: The authors argue that because in the low-latitude heating case the vertical heat transport is more efficient, the radiative cooling to space is also more efficient, providing a negative feedback to the surface warming. It's not obvious to me why. I guess the authors are implying a lapse rate feedback. Are you sure that with the gray radiation scheme in this model and in the absence of clouds the lapse rate feedback is still valid? Maybe, but I don't think it's trivial. If the authors use this argument, I think it requires some more explanation.

**9. Although the gray radiation model has no cloud, the model still has prognostic water vapour that can evaporate from the surface, condense and immediately precipitate out in the troposphere. Although the radiation in the model does not interact with the water vapour, the latent heat released during condensation can still warm the troposphere and change the lapse rate. Previous studies show that the lapse rate feedback is important to shape the temperature response in such a gray radiation model (e.g., Henry and Merlis 2019; Feldl and Merlis 2021).**

5) In the same paragraph change "balanc" to "balance".

**10. Thank you and we have now made the changes. Please see line 254.**

References:

Henry, M. and Merlis, T. M.: The role of the nonlinearity of the Stefan–Boltzmann law on the structure of radiatively forced temperature change, Journal of Climate, 32, 335–348, https://doi.org/10.1175/JCLI-D-17-0603.1, 2019.

Feldl, N. and Merlis, T. M.: Polar amplification in idealized climates: The role of ice, moisture, and seasons, Geophysical Research Letters, 48, e2021GL094 130, https://doi.org/10.1029/2021GL094130, 2021.